# Compute- and Memory-Efficient Reinforcement Learning with Latent Experience Replay

## Abstract

Recent advances in off-policy deep reinforcement learning (RL) have led to impressive success in complex tasks from visual observations. Experience replay improves sample-efficiency by reusing experiences from the past, and convolutional neural networks (CNNs) process high-dimensional inputs effectively. However, such techniques demand high memory and computational bandwidth. In this paper, we present Latent Vector Experience Replay (LeVER), a simple modification of existing off-policy RL methods, to address these computational and memory requirements without sacrificing the performance of RL agents. To reduce the computational overhead of gradient updates in CNNs, we freeze the lower layers of CNN encoders early in training due to early convergence of their parameters. Additionally, we reduce memory requirements by storing the low-dimensional latent vectors for experience replay instead of high-dimensional images, enabling an adaptive increase in the replay buffer capacity, a useful technique in constrained-memory settings. In our experiments, we show that LeVER does not degrade the performance of RL agents while significantly saving computation and memory across a diverse set of DeepMind Control environments and Atari games. Finally, we show that LeVER is useful for computation-efficient transfer learning in RL because lower layers of CNNs extract generalizable features, which can be used for different tasks and domains.

## 1 Introduction

Success stories of deep reinforcement learning (RL) from high dimensional inputs such as pixels or large spatial layouts include achieving superhuman performance on Atari games (Mnih et al., 2015; Schrittwieser et al., 2019; Badia et al., 2020), grandmaster level in Starcraft II (Vinyals et al., 2019) and grasping a diverse set of objects with impressive success rates and generalization with robots in the real world (Kalashnikov et al., 2018). Modern off-policy RL algorithms (Mnih et al., 2015; Hessel et al., 2018; Hafner et al., 2019; 2020; Srinivas et al., 2020; Kostrikov et al., 2020; Laskin et al., 2020) have improved the sample-efficiency of agents that process high-dimensional pixel inputs with convolutional neural networks (CNNs; LeCun et al. 1998) using the past experiential data that is typically stored as raw observations form in a replay buffer (Lin, 1992). However, these methods demand high memory and computational bandwidth, which makes deep RL inaccessible in several scenarios, such as learning with much lighter on-device computation (e.g. mobile phones or other light-weight edge devices).

For compute- and memory-efficient deep learning, several strategies, such as network pruning (Han et al., 2015; Frankle & Carbin, 2019), quantization (Han et al., 2015; Iandola et al., 2016) and freezing (Yosinski et al., 2014; Raghu et al., 2017) have been proposed in supervised learning and unsupervised learning for various purposes (see Section 2 for more details). In computer vision, Raghu et al. (2017) showed that the computational cost of updating CNNs can be reduced by freezing lower layers earlier in training, and Han et al. (2015) introduced a deep compression, which reduces the memory requirement of neural networks by producing a sparse network. In natural language processing, several approaches (Tay et al., 2019; Sun et al., 2020) have studied improving the computational efficiency of Transformers (Vaswani et al., 2017). In deep RL, however, developing compute- and memory-efficient techniques has received relatively little attention despite their serious impact on the practicality of RL algorithms.

In this paper, we propose **L**atent **V**ector **E**xperience **R**eplay (LeVER), a simple technique to reduce computational overhead and memory requirements that is compatible with various off-policy RL algorithms (Haarnoja et al., 2018; Hessel et al., 2018; Srinivas et al., 2020). Our main idea is to freeze the lower layers of CNN encoders of RL agents early in training, which enables two key capabilities: (a) compute-efficiency: reducing the computational overhead of gradient updates in CNNs; (b) memory-efficiency: saving memory by storing the low-dimensional latent vectors to experience replay instead of high-dimensional images. Additionally, we leverage the memory-efficiency of LeVER to adaptively increase the replay capacity, resulting in improved sample-efficiency of off-policy RL algorithms in constrained-memory settings. LeVER achieves these improvements without sacrificing the performance of RL agents due to early convergence of CNN encoders.

To summarize, the main contributions of this paper are as follows:

- We present LeVER, a compute- and memory-efficient technique that can be used in conjunction with most modern off-policy RL algorithms (Haarnoja et al., 2018; Hessel et al., 2018).
- We show that LeVER significantly reduces computation while matching the original performance of existing RL algorithms on both continuous control tasks from DeepMind Control Suite (Tassa et al., 2018) and discrete control tasks from Atari games (Bellemare et al., 2013).
- We show that LeVER improves the sample-efficiency of RL agents in constrained-memory settings by enabling an increased replay buffer capacity.
- Finally, we show that LeVER is useful for computation-efficient transfer learning, highlighting the generality and transferability of encoder features.

## 2 RELATED WORK

**Off-policy deep reinforcement learning.** The most sample-efficient RL agents often use off-policy RL algorithms, a recipe for improving the agent's policy from experiences that may have been recorded with a different policy (Sutton & Barto, 2018). Off-policy RL algorithms are typically based on Q-Learning (Watkins & Dayan, 1992) which estimates the optimal value functions for the task at hand, while actor-critic based off-policy methods (Lillicrap et al., 2016; Schulman et al., 2017; Haarnoja et al., 2018) are also commonly used. In this paper we will consider Deep Q-Networks (DQN; Mnih et al. 2015),which combine the function approximation capability of deep convolutional neural networks (CNNs; LeCun et al. 1998) with Q-Learning along with the usage of the experience replay buffer (Lin, 1992) as well as off-policy actor-critic methods (Lillicrap et al., 2016; Haarnoja et al., 2018), which have been proposed for continuous control tasks.

Taking into account the learning ability of humans and practical limitations of wall clock time for deploying RL algorithms in the real world, particularly those that learn from raw high dimensional inputs such as pixels (Kalashnikov et al., 2018), the sample-inefficiency of off-policy RL algorithms has been a research topic of wide interest and importance (Lake et al., 2017; Kaiser et al., 2020). To address this, several improvements in pixel-based off-policy RL have been proposed recently: algorithmic improvements such as Rainbow (Hessel et al., 2018) and its data-efficient versions (van Hasselt et al., 2019); using ensemble approaches based on bootstrapping (Osband et al., 2016; Lee et al., 2020); combining RL algorithms with auxiliary predictive, reconstruction and contrastive losses (Jaderberg et al., 2017; Higgins et al., 2017; Oord et al., 2018; Yarats et al., 2019; Srinivas et al., 2020; Stooke et al., 2020); using world models for auxiliary losses and/or synthetic rollouts (Sutton, 1991; Ha & Schmidhuber, 2018; Kaiser et al., 2020; Hafner et al., 2020); using data-augmentations on images to improve sample-efficiency (Laskin et al., 2020; Kostrikov et al., 2020).

**Compute-efficient techniques in machine learning.** Most recent progress in deep learning and RL has relied heavily on the increased access to more powerful computational resources. To address this, Mattson et al. (2020) presented MLPerf, a fair and precise ML benchmark to evaluate model training time on standard datasets, driving scalability alongside performance, following a recent focus on mitigating the computational cost of training ML models. Several techniques, such as pruning and quantization (Han et al., 2015; Frankle & Carbin, 2019; Blalock et al., 2020; Iandola et al., 2016; Tay et al., 2019) have been developed to address compute and memory requirements. Raghu et al. (2017) proposed freezing earlier layers to remove computationally expensive backward passes in supervised learning tasks, motivated by the bottom-up convergence of neural networks. This intuition was

further extended to recurrent neural networks (Morcos et al., 2018) and continual learning (Pellegrini et al., 2019), and Yosinski et al. (2014) study the transferability of frozen and fine-tuned CNN parameters. Fang et al. (2019) store low-dimensional embeddings of input observations in scene memory for long-horizon tasks. We focus on the feasibility of freezing neural network layers in deep RL and show that this idea can improve the compute- and memory-efficiency of many off-policy algorithms using standard RL benchmarks.

## 3  BACKGROUND

We formulate visual control task as a partially observable Markov decision process (POMDP; Sutton & Barto 2018; Kaelbling et al. 1998). Formally, at each timestep $t$, the agent receives a high-dimensional observation $o_t$, which is an indirect representation of the state $s_t$, and chooses an action $a_t$ based on its policy $\pi$. The environment returns a reward $r_t$ and the agent transitions to the next observation $o_{t+1}$. The return $R_t = \sum_{k=0}^{\infty} \gamma^k r_{t+k}$ is the total accumulated rewards from timestep $t$ with a discount factor $\gamma \in [0, 1)$. The goal of RL is to learn a policy $\pi$ that maximizes the expected return over trajectories. By following the common practice in DQN (Mnih et al., 2015), we handle the partial observability of environment using stacked input observations, which are processed through the convolutional layers of an encoder $f_\psi$.

**Soft Actor-Critic**. SAC (Haarnoja et al., 2018) is an off-policy actor-critic method based on the maximum entropy RL framework (Ziebart, 2010), which encourages the robustness to noise and exploration by maximizing a weighted objective of the reward and the policy entropy. To update the parameters, SAC alternates between a soft policy evaluation and a soft policy improvement. At the soft policy evaluation step, a soft Q-function, which is modeled as a neural network with parameters $\theta$, is updated by minimizing the following soft Bellman residual:

$$\mathcal{L}_Q^{\mathtt{SAC}}(\theta, \psi) = \mathbb{E}_{\tau_t \sim \mathcal{B}}\Bigg[\bigg(Q_\theta(f_\psi(o_t), a_t) - r_t$$
$$- \gamma \mathbb{E}_{a_{t+1} \sim \pi_\phi}\big[Q_{\bar{\theta}}(f_{\bar{\psi}}(o_{t+1}), a_{t+1}) - \alpha \log \pi_\phi(a_{t+1}|f_\psi(o_{t+1}))\big]\bigg)^2\Bigg],$$

where $\tau_t = (o_t, a_t, r_t, o_{t+1})$ is a transition, $\mathcal{B}$ is a replay buffer, $\bar{\theta}, \bar{\psi}$ are the delayed parameters, and $\alpha$ is a temperature parameter. At the soft policy improvement step, the policy $\pi$ with its parameter $\phi$ is updated by minimizing the following objective:

$$\mathcal{L}_\pi^{\mathtt{SAC}}(\phi) = \mathbb{E}_{o_t \sim \mathcal{B}, a_t \sim \pi_\phi}\big[\alpha \log \pi_\phi(a_t|f_\psi(o_t)) - Q_\theta(f_\psi(o_t), a_t)\big]. \tag{1}$$

Here, the policy is modeled as a Gaussian with mean and covariance given by neural networks to handle continuous action spaces.

**Deep Q-learning.** DQN algorithm (Mnih et al., 2015) learns a Q-function, which is modeled as a neural network with parameters $\theta$, by minimizing the following Bellman residual:

$$\mathcal{L}^{\mathtt{DQN}}(\theta, \psi) = \mathbb{E}_{\tau_t \sim \mathcal{B}}\Bigg[\bigg(Q_\theta(f_\psi(o_t), a_t) - r_t - \gamma \max_a Q_{\bar{\theta}}(f_{\bar{\psi}}(o_{t+1}), a)\bigg)^2\Bigg], \tag{2}$$

where $\tau_t = (o_t, a_t, r_t, o_{t+1})$ is a transition, $\mathcal{B}$ is a replay buffer, and $\bar{\theta}, \bar{\psi}$ are the delayed parameters. Rainbow DQN integrates several techniques, such as double Q-learning (Van Hasselt et al., 2016) and distributional DQN (Bellemare et al., 2017). For exposition, we refer the reader to Hessel et al. (2018) for more detailed explanations of Rainbow DQN.

## 4  LEVER: LATENT VECTOR EXPERIENCE REPLAY

In this section, we present LeVER: **Le**tent **V**ector **E**xperience **R**eplay, which can be used in conjunction with most modern off-policy RL algorithms, such as SAC (Haarnoja et al., 2018) and Rainbow DQN (Hessel et al., 2018). Our main idea is to freeze lower layers during training and only update higher layers, which eliminates the computational overhead of computing gradients and updating in lower layers. We additionally improve the memory-efficiency of off-policy RL algorithms by storing low-dimensional latent vectors in the replay buffer instead of high-dimensional pixel observations. See Figure 1 and Appendix A for more details of our method.

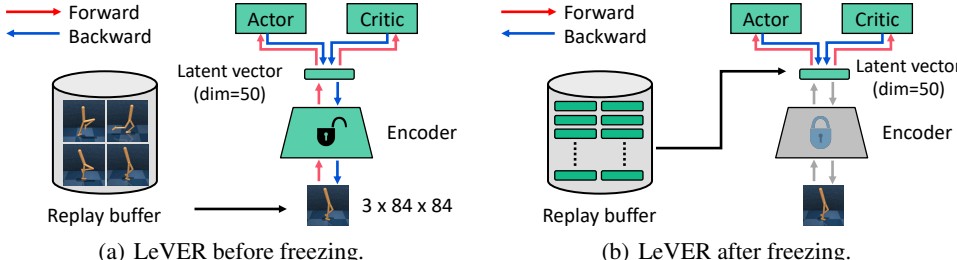

(a) LeVER before freezing.     (b) LeVER after freezing.

Figure 1: Illustration of our framework. (a) Before the encoder is frozen, all forward and backward passes are active through the network, and we store images in the replay buffer. (b) After freezing, we store latent vectors in the replay buffer, and remove all forward and backward passes through the encoder. We remark that more samples can be stored in the replay buffer due to the relatively low dimensionality of the latent vector.

## 4.1 FREEZING ENCODER FOR SAVING COMPUTATION AND MEMORY

We process high-dimensional image input with an encoder $f_\psi$ to obtain $z_t = f_\psi(o_t)$, which is used as input for policy $\pi_\phi$ and Q-function $Q_\theta$ as described in Section 3. In off-policy RL, we store transitions $(o_t, a_t, o_{t+1}, r_t)$ in the replay buffer $\mathcal{B}$ to improve sample-efficiency by reusing experience from the past. However, processing high-dimensional image input $o_t$ is computationally expensive. To handle this issue, after $T_f$ updates, we freeze the parameters of encoder $\psi$, and only update the policy and Q-function. We remark that this simple technique can save computation without performance degradation because the encoder is modeled as deep convolutional neural networks, while a shallow MLP is used for policy and Q-function. Freezing lower layers of neural networks also has been investigated in supervised learning based on the observation that neural networks converge to their final representations *from the bottom-up*, i.e., lower layers converge very early in training (Raghu et al., 2017). For the first time, we show the feasibility and effectiveness of this idea in pixel-based reinforcement learning (see Figure 7(a) for supporting experimental results) and present solutions to its RL-specific implementation challenges.

Moreover, in order to save memory, we consider storing (compressed) latent vectors instead of high-dimensional image inputs. Specifically, each experience in $\mathcal{B}$ is replaced by the latent transition $(z_t, a_t, z_{t+1}, r_t)$, and the replay capacity is increased to $\widehat{C}$ (see Section 4.2 for more details). Thereafter, for each subsequent environment interaction, the latent vectors $z_t = f_\psi(o_t)$ and $z_{t+1} = f_\psi(o_{t+1})$ are computed prior to storing $(z_t, a_t, z_{t+1}, r_t)$ in $\mathcal{B}$. During agent updates, the sampled latent vectors are directly passed into the policy $\pi_\phi$ and Q-function $Q_\theta$, bypassing the encoder convolutional layers. Since the agent samples and trains with latent vectors after freezing, we only store the latent vectors and avoid the need to maintain large image observations in $\mathcal{B}$.

## 4.2 ADDITIONAL TECHNIQUES AND DETAILS FOR LEVER

**Data augmentations.** Recently, various data augmentations (Srinivas et al., 2020; Laskin et al., 2020; Kostrikov et al., 2020) have provided large gains in the sample-efficiency of RL from pixel observations. However, LeVER precludes data augmentations because we store the latent vector instead of the raw pixel observation. We find that the absence of data augmentations could decrease sample-efficiency in some cases, e.g., when the capacity of $\mathcal{B}$ is small. To mitigate this issue, we perform $K$ number of different data augmentations for each input observation $o_t$ and store $K$ distinct latent vectors $\{z_t^k = f_\psi(\text{AUG}_k(o_t)) | k = 1 \cdots K\}$. We find empirically that $K = 4$ achieves competitive performance to standard RL algorithms in most cases.

**Increasing replay capacity.** By storing the latent vector in replay buffer, we can adaptively increase the capacity (i.e., total number of transitions), which is determined by the size difference between the input pixel observations and the latent vectors output by the encoder, with a few additional considerations. The new capacity of the replay buffer is

$$\widehat{C} = \left\lfloor C * \left( \tfrac{P}{4NKL} \right) \right\rfloor,$$

where $C$ is the capacity of the original replay buffer, $P$ is the size of the raw observation, $L$ is the size of the latent vector, and $K$ is the number of data augmentations. The number of encoders $N$ is algorithm-specific and determines the number of distinct latent vectors encountered for each observation during training. For Q-learning algorithms $N = 1$, whereas for actor-critic algorithms $N = 2$ if the actor and critic each compute their own latent vectors. Some algorithms employ a target network for updating the Q-function (Mnih et al., 2015; Haarnoja et al., 2018), but we use the same latent vectors for the online and target networks after freezing to avoid storing target latent vectors separately and find that tying their parameters does not degrade performance.[1] The factor of 4 arises from the cost of saving floats for latent vectors, while raw pixel observations are saved as integer pixel values. We assume the memory required for actions, and rewards is small and only consider only the memory used for observations.

## 5 EXPERIMENTAL RESULTS

We designed our experiments to answer the following questions:

- Can LeVER reduce the computational overhead of various off-policy RL algorithms for both continuous (see Figure 2) and discrete (see Figure 3) control tasks?
- Can LeVER reduce the memory consumption and improve the sample-efficiency of off-policy RL algorithms by adaptively increasing the buffer size (see Figure 4 and Figure 5)?
- Can LeVER be useful for computation-efficient transfer learning (see Figure 7(a))?
- Do CNN encoders of RL agents converge early in training (see Figure 7(b) and Figure 7(c))?

### 5.1 SETUPS

**Computational efficiency.** We first demonstrate the computational efficiency of LeVER on the DeepMind Control Suite (DMControl; Tassa et al. 2018) and Atari games (Bellemare et al., 2013) benchmarks. DMControl is commonly used for benchmarking sample-efficiency for image-based continuous control methods. For DMControl experiments, we consider a state-of-the-art model-free RL method, which applies contrastive learning (CURL; Srinivas et al. 2020) to SAC (Haarnoja et al., 2018), using the image encoder architecture from SAC-AE (Yarats et al., 2019). For evaluation, we compare the computational efficiency of CURL with and without LeVER by measuring floating point operations (FLOPs).[2] For discrete control tasks from Atari games, we perform similar experiments comparing the FLOPs required by Rainbow (Hessel et al., 2018) with and without LeVER. For both our method and the baseline, we use the hyperparameters and encoder architecture of data-efficient Rainbow (van Hasselt et al., 2019). We train our method and the baseline for 500K timesteps as done in Srinivas et al. (2020) and Laskin et al. (2020).

**Memory efficiency.** We showcase the memory efficiency of LeVER with a set of constrained-memory experiments in DMControl. For Cartpole and Finger, the memory allocated for storing observations is constrained to 0.03 GB, corresponding to an initial replay buffer capacity $C = 1000$. For Reacher and Walker, the memory is constrained to 0.06 GB for an initial capacity of $C = 2000$ due to the difficulty of learning in these environments. In this constrained-memory setting, we compare the sample-efficiency of CURL with and without LeVER. As an upper bound, we also report the performance of CURL without memory constraints, i.e., the replay capacity is set to the number of training steps. For Atari experiments, the baseline agent is data-efficient Rainbow and the memory allocation is 0.07 GB, corresponding to initial replay capacity $C = 10000$. The other hyperparameters are the same as those in the computational efficiency experiments.

The encoder architecture used for our experiments with CURL is used in Yarats et al. (2019). It consists of four convolutional layers with 3 x 3 kernels and 32 channels, with the ReLU activation applied after each conv layer. The architecture used for our Rainbow experiments is from van Hasselt et al. (2019), consisting of a convolutional layer with 32 channels followed by a convolutional layer with 64 channels, both with 5 x 5 kernels and followed by a ReLU activation. For our method, we freeze the first fully-connected layer of the actor and critic in CURL experiments and

---

[1] We remark that the higher layers of the target network are not tied to the online network after freezing.

[2] We explain our procedure for counting the number of FLOPs in Appendix B.

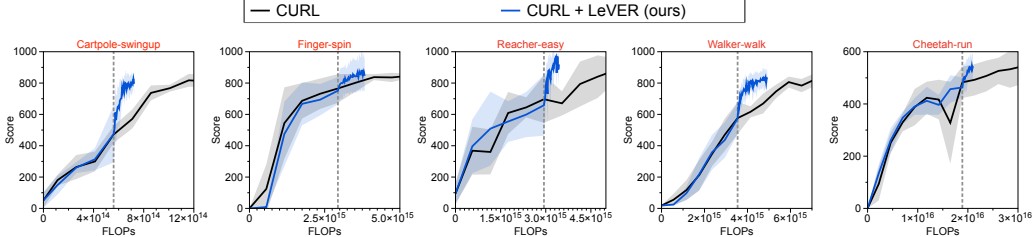

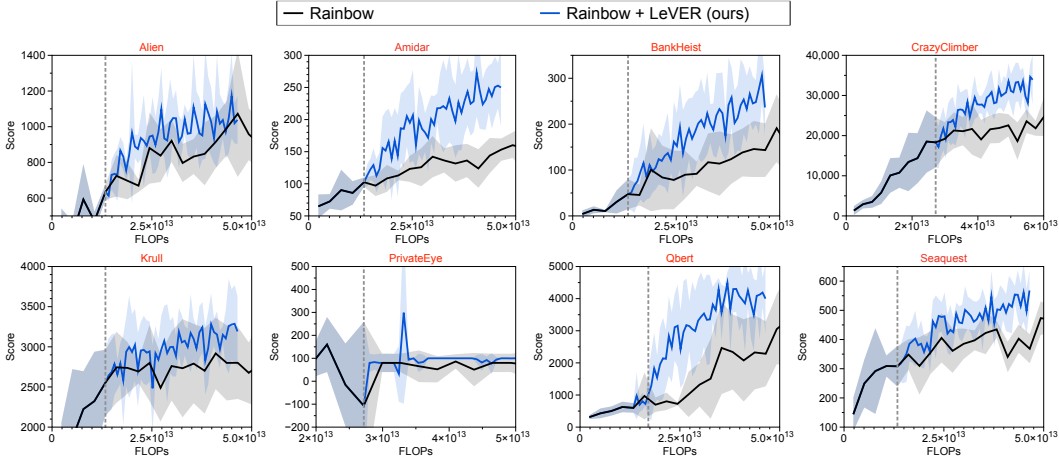

Figure 2: Learning curves for CURL with and without LeVER, where the x-axis shows FLOPs. The dotted gray line denotes the encoder freezing time $t = T_f$. The solid line and shaded regions represent the mean and standard deviation, respectively, across five runs.

Figure 3: Learning curves for Rainbow with and without LeVER, where the x-axis shows FLOPs. The dotted gray line denotes the encoder freezing time $t = T_f$. The solid line and shaded regions represent the mean and standard deviation, respectively, across five runs.

the last convolutional layer of the encoder in Rainbow experiments (see Appendix E for justification). We present the best results achieved by LeVER across various values of $T_f$, the number of training steps before freezing the encoder. The full list of hyperparameters is provided in Appendix G (DMControl) and Appendix H (Atari).

## 5.2 IMPROVING COMPUTE- AND MEMORY-EFFICIENCY

Experimental results in DMControl and Atari showcasing the computational efficiency of LeVER are provided in Figures 2 and Figure 3. CURL and Rainbow both achieve higher performance within significantly fewer FLOPs when combined with LeVER in DMControl and Atari, respectively. Additionally, Table 1 compares the performance of Rainbow with and without LeVER at 45T (4.5e13) FLOPs. In particular, the average returns are improved from 145.8 to 276.6 compared to baseline Rainbow in BankHeist and from 2325.5 to 4123.5 in Qbert. We remark that LeVER achieves better computational efficiency while maintaining the agent's final performance and comparable sample-efficiency (see Appendix E for corresponding figures).

Experimental results in Atari and DMControl showcasing the sample-efficiency of LeVER in the constrained-memory setup are provided in Figure 4 and Figure 5. CURL and Rainbow achieve higher final performance and better sample-efficiency when combined with LeVER in DMControl and Atari, respectively. Additionally, Table 1 compares the performance of unbounded memory Rainbow and constrained-memory (0.07 GB) Rainbow with and without LeVER at 500K environment interactions. In particular, the average returns are improved from 10498.0 to 17620.0 compared to baseline Rainbow in CrazyClimber and from 2430.5 to 3231.0 in Qbert. Although we disentangle the computational and memory benefits of LeVER in these experiments, we also highlight the

|  | Scores at 45T FLOPs | | Scores at 500K environment steps (0.07GB) | |
| --- | --- | --- | --- | --- |
|  | Rainbow | Rainbow+LeVER | Rainbow | Rainbow+LeVER |
| Alien | $992.0 \pm 152.7$ | **1172.6** ±239.0 | $1038.4 \pm 101.1$ | **1134.6** ±452.9 |
| Amidar | $144.0 \pm 27.4$ | **250.5** ±47.4 | $121.0 \pm 31.2$ | **165.3** ±47.6 |
| BankHeist | $145.8 \pm 61.2$ | **276.6** ±98.1 | **161.6** ±57.7 | $151.8 \pm 65.8$ |
| CrazyClimber | $21580.0 \pm 3514.6$ | **28066.0** ±4108.5 | $10498.0 \pm 1387.8$ | **17620.0** ±4418.4 |
| Krull | $2799.5 \pm 468.1$ | **3277.5** ±440.5 | $2215.7 \pm 336.9$ | **3069.2** ±377.6 |
| PrivateEye | $81.5 \pm 37.0$ | **100.0** ±0.0 | **80.0** ±40.0 | **80.0** ±40.0 |
| Qbert | $2325.5 \pm 1152.7$ | **4123.5** ±1385.5 | $2430.5 \pm 658.8$ | **3231.0** ±1567.6 |
| Seaquest | $402.8 \pm 48.4$ | **561.2** ±100.5 | $262.8 \pm 19.1$ | **336.8** ±45.9 |

Table 1: Scores on Atari games at 45T FLOPs corresponding to Figure 3 and scores on Atari games at 500K environment interactions in the constrained-memory setup (0.07GB) corresponding to Figure 4. The results show the mean and standard deviation averaged five runs, and the best results are indicated in bold.

computational gain of LeVER in constrained-memory settings (effectively combining the benefits) in Appendix D.

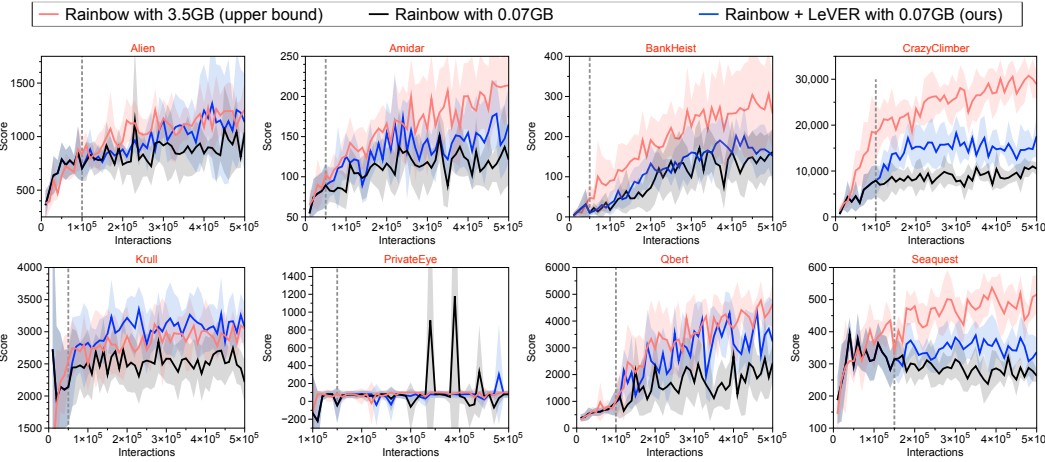

Figure 4: Comparison of the sample-efficiency of Rainbow with and without LeVER in constrained-memory (0.07 GB) settings. The dotted gray line denotes the encoder freezing time $t = T_f$. The solid line and shaded regions represent the mean and standard deviation, respectively, across five runs.

## 5.3 FREEZING LARGER CONVOLUTIONAL ENCODERS

We also verify the benefits of LeVER using deeper convolutional encoders, which are widely used in a range of applications such as visual navigation tasks and favored for their superior generalization ability. Specifically, we follow the setup described in Section 5.1 and replace the SAC-AE architecture (4 convolutional layers) with the IMPALA architecture (Espeholt et al., 2018) (15 convolutional layers containing residual blocks (He et al., 2016)). Figure 6(b) shows the computational efficiency of LeVER in Cartpole-swingup and Walker-walk with the IMPALA architecture. CURL achieves higher performance within significantly fewer FLOPs when combined with LeVER. We remark that the gains due to LeVER are more significant because computing and updating gradients for large convolutional encoders is very computationally expensive.

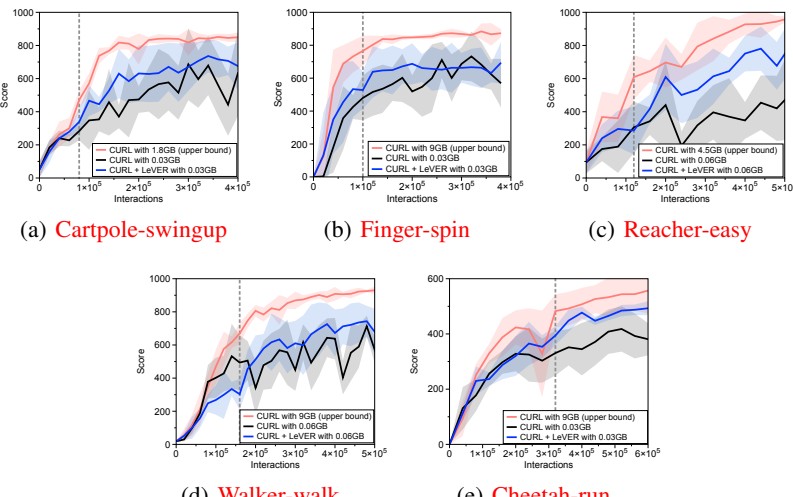

Figure 5: Comparison of the sample-efficiency of CURL with and without LeVER in constrained-memory settings. The dotted gray line denotes the encoder freezing time $t = T_f$. The solid line and shaded regions represent the mean and standard deviation, respectively, across five runs.

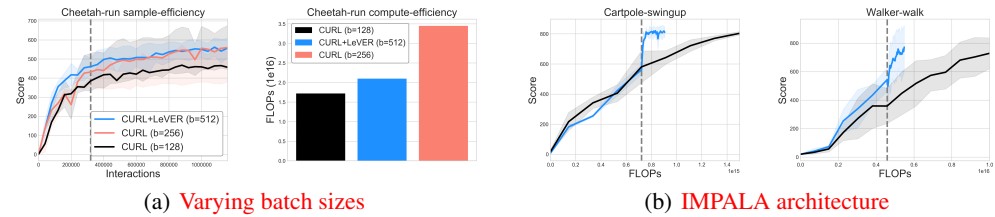

Figure 6: (a) Left: Cheetah-run learning curves for CURL with and without LeVER, with batch sizes b=512 and b=128, respectively, where the x-axis shows samples. Right: Number of FLOPs used by each agent to achieve its final performance. (b) Learning curves for CURL using the IMPALA architecture with and without LeVER, where the x-axis shows FLOPs. The dotted gray line denotes the encoder freezing time $t = T_f$. The solid line and shaded regions represent the mean and standard deviation, respectively, across five runs.

## 5.4 IMPROVING SAMPLE EFFICIENCY WITH LeVER AND LARGER BATCH SIZES

In this subsection we show that we can combine LeVER with larger batch sizes to improve the sample-efficiency of RL agents. Larger batch sizes have been shown to improve agent performance in many settings, but require more compute since each gradient calculation and update is performed on more observations. We demonstrate that LeVER can mitigate these issues by showing results in Cheetah-run, a task known to achieve better performance with larger batch sizes. Figure 6(a) shows the sample-efficiency of CURL (batch 128) and CURL+LeVER (batch 512), and the corresponding computational efficiency of each agent. CURL achieves better sample-efficiency when combined with LeVER and the larger batch size, but does this within a comparable compute budget. In contrast, CURL (batch 256) requires significantly more compute to achieve similar performance to CURL+LeVER (batch 512).

## 5.5 IMPROVING COMPUTATIONAL EFFICIENCY IN TRANSFER SETTINGS

We demonstrate, as another application of our method, that LeVER increases computational efficiency in the transfer setting: utilizing the parameters from Task A on unseen Tasks B. Specifically, we train a CURL agent for 60K environment interactions on Walker-stand; then, we only fine-tune

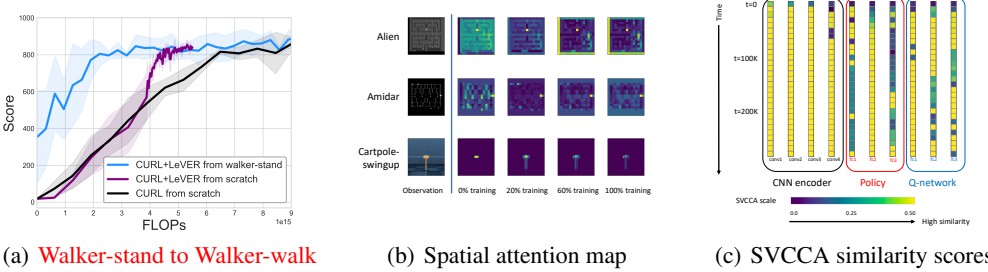

(a) Walker-stand to Walker-walk        (b) Spatial attention map        (c) SVCCA similarity scores

Figure 7: (a) Comparison of the computational efficiency of agents trained from scratch with CURL and agents trained with CURL+LeVER from Walker-stand pretraining. (b) Spatial attention map from CNN encoders. (c) SVCCA (Raghu et al., 2017) similarity scores between each layer and itself at time t and t+10K throughout training for Walker-walk.

the policy and Q-functions on unseen tasks (e.g., Walker-walk and Cheetah-run) using network parameters from Walker-stand. To save computation, during fine-tuning, we freeze the encoder parameters. Figure 7(a) shows the computational gain of LeVER in task transfer (i.e., Walker-stand to Walker-walk similar to Yarats et al. (2019)), and domain transfer (i.e., Walker-stand to Cheetah-run and Walker-stand to Hopper-hop) is shown in Appendix C. Due to the generality of CNN features, we can achieve this computational gain using a pretrained encoder. For the task transfer setup, we provide more analysis on the number of frozen layers and freezing time hyperparameter $T_f$ in Appendix C.

## 5.6 ENCODER ANALYSIS

In this subsection we present visualizations to verify that the neural networks employed in deep reinforcement learning indeed converge *from the bottom up*, similar to those used in supervised learning (Raghu et al., 2017). Figure 7(b) shows the spatial attention map for two Atari games and one DMControl environment at various points during training. Similar to Laskin et al. (2020) and Zagoruyko & Komodakis (2017), we compute the spatial attention map by mean-pooling the absolute values of the activations along the channel dimension and follow with a 2-dimensional spatial softmax. The attention map shows significant change in the first 20% of training, and remains relatively unchanged thereafter, suggesting that the encoder converges to its final representations early in training. Figure 7(c) shows the SVCCA (Raghu et al., 2017) score, a measure of neural network layer similarity, between a layer and itself at time t and t+10K. The convolutional layers of the encoder achieve high similarity scores with themselves between time t and t+10K, while the higher layers of the policy and Q-network continue to change throughout training. In our DMControl environments we freeze the convolutional layers and the first fully-connected layer of the policy and Q-network (denoted fc1). Although the policy fc1 continues to change, the convergence of the Q-network fc1 and the encoder layers allow us to achieve our computational and memory savings with minimal performance degradation.

## 6 CONCLUSION

In this paper, we presented LeVER, a simple but powerful modification of off-policy RL algorithms that significantly reduces computation and memory requirements while maintaining state-of-the-art performance. We leveraged the intuition that CNN encoders in deep RL converge to their final representations early in training to freeze the encoder and subsequently store latent vectors to save computation and memory. In our experimental results, we demonstrated the compute- and memory-efficiency of LeVER in various DMControl environments and Atari games, and proposed a technique for computation-efficient transfer learning. With LeVER, we highlight the potential for improvements in compute- and memory-efficiency in deep RL that can be made without sacrificing performance, in hopes of making deep RL more practical and accessible in the real world.

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

# Appendix

## A  ALGORITHM

We detail the specifics of modifying off-policy RL methods with LeVER below. For concreteness, we describe LeVER combined with deep Q-learning methods.

---

**Algorithm 1** Latent Vector Experience Replay (DQN Base Agent)

---

1: Initialize replay buffer $\mathcal{B}$ with capacity $C$
2: Initialize action-value network $Q$ with parameters $\theta$ and encoder $f$ with parameters $\psi$
3: **for** each timestep $t$ **do**
4:     Select action: $a_t \leftarrow \arg\max_a Q_\theta(f_\psi(o_t), a)$
5:     Collect observation $o_{t+1}$ and reward $r_t$ from the environment by taking action $a_t$
6:     **if** $t \leq T_f$ **then**
7:         Store transition $(o_t, a_t, o_{t+1}, r_t)$ in replay buffer $\mathcal{B}$
8:     **else**
9:         Compute latent states $z_t, z_{t+1} \leftarrow f_\psi(o_t), f_\psi(o_{t+1})$
10:         Store transition $(z_t, a_t, z_{t+1}, r_t)$ in replay buffer $\mathcal{B}$
11:     **end if**
12:     // REPLACE PIXEL-BASED TRANSITIONS WITH LATENT TRAJECTORIES
13:     **if** $t = T_f$ **then**
14:         Compute latent states $\{(z_t, z_{t+1})\}_{t=1}^{\min(T_f, c)} \leftarrow \{(f_\psi(o_t), f_\psi(o_{t+1}))\}_{t=1}^{\min(T_f, c)}$
15:         Replace $\{(o_t, a_t, o_{t+1}, r_t)\}_{t=1}^{\min(T_f, c)}$ with latent transitions $\{(z_t, a_t, z_{t+1}, r_t)\}_{t=1}^{\min(T_f, c)}$
16:         Increase the capacity of $\mathcal{B}$ to $\widehat{C}$
17:     **end if**
18:     // UPDATE PARAMETERS OF Q-NETWORK WITH SAMPLED IMAGES OR LATENTS
19:     **for** each gradient step **do**
20:         **if** $t < T_f$ **then**
21:             Sample random minibatch $\{(o_j, a_j, o_{j+1}, r_j)\}_{j=1}^b \sim \mathcal{B}$
22:             Calculate target $y_j = r_j + \gamma \max_{a'} Q_{\bar{\theta}}(f_{\bar{\psi}}(o_{j+1}), a')$
23:             Perform a gradient step on $\mathcal{L}^{\texttt{DQN}}(\theta, \psi)$ (2)
24:         **else**
25:             Sample random minibatch $\{(z_j, a_j, z_{j+1}, r_j)\}_{j=1}^b \sim \mathcal{B}$
26:             Calculate target $y_j = r_j + \gamma \max_{a'} Q_{\bar{\theta}}(z_{j+1}, a')$
27:             Perform a gradient step on $\mathcal{L}^{\texttt{DQN}}(\theta)$ (2)
28:         **end if**
29:     **end for**
30: **end for**

---

## B  CALCULATION OF FLOATING POINT OPERATIONS

We consider each backward pass to require twice as many FLOPs as a forward pass. [3] Each weight requires one multiply-add operation in the forward pass. In the backward pass, it requires two multiply-add operations: at layer $i$, the gradient of the loss with respect to the weight at layer $i$ and with respect to the output of layer $(i-1)$ need to be computed. The latter computation is necessary for subsequent gradient calculations for weights at layer $(i-1)$.

We use functions from Huang et al. (2018) and Jeong & Shin (2019) to obtain the number of operations per forward pass for all layers in the encoder (denoted $E$) and number of operations per forward pass for all MLP layers (denoted $M$). For concreteness, we provide a FLOPs breakdown by layer for the architectures we use in Table 2 and 3.

---

[3]This method for FLOP calculation is used in https://openai.com/blog/ai-and-compute/.

We denote the number of forward passes per iteration $F$, the number of backward passes per iteration $B$, and the batch size $b$. We assume the number of updates per timestep is 1. Then, the number of FLOPs per iteration before freezing at time $t = T_f$ is:

$$bF(E + M) + 2bB(E + M)$$

For the baseline, FLOPs are computed using this formula throughout training.

LeVER reduces computational overhead by eliminating most of the encoder forward and backward passes. The number of FLOPs per iteration after freezing is:

$$bFM + 2bBM + EKN$$

where $K$ is the number of data augmentations and $N$ is the number of networks as described in Section 4.2. The forward and backward passes of the encoder are removed, with the exception of the $EKN$ term at the end that arises from calculating latent vectors for the current observation.

At freezing time $t = T_f$, we need to compute latent vectors for each transition in the replay buffer. This introduces a one-time cost of $(EKN \min(T_f, C))$ FLOPs, since the number of transitions in the replay buffer is $\min(T_f, C)$, where $C$ is the initial replay capacity.

Table 2: Forward pass FLOPs breakdown by layer for DM Control experiments.

| Layer | FLOPs | Layer | FLOPs | Layer | FLOPs |
|---|---|---|---|---|---|
| encoder:conv1 | 4.3e6 | critic:fc1 | 1.9e6 | actor:fc1 | 1.9e6 |
| encoder:conv2 | 1.4e7 | critic:fc2 | 5.3e4 | actor:fc2 | 5.2e4 |
| encoder:conv3 | 1.3e7 | critic:fc3 | 1.0e6 | actor:fc3 | 1.0e6 |
| encoder:conv4 | 1.1e7 | critic:fc4 | 1.0e3 | actor:fc4 | 2.0e3 |

Table 3: Forward pass FLOPs breakdown by layer for Atari experiments.

| Layer | FLOPs | Layer | FLOPs | Layer | FLOPs |
|---|---|---|---|---|---|
| encoder:conv1 | 8.2e5 | value stream:fc1 | 1.5e5 | advantage stream:fc1 | 1.5e5 |
| encoder:conv2 | 4.6e5 | value stream:fc2 | 1.3e4 | advantage stream:fc2 | 1.2e5 |

## C  ADDITIONAL TRANSFER EXPERIMENTS

**Domain transfer.** In Figure 7(a) we show the computational efficiency of LeVER in a task transfer setting. Here we show in Figure 8 that frozen encoder parameters can also be used in domain transfer tasks (i.e. Walker-stand to Cheetah-run and Walker-stand to Hopper-hop). In this setting, we only transfer the encoder parameters, whereas we transfer the entire network in task transfer.

**Transfer setting analysis.** In Figure 7(a) we show the computational efficiency of LeVER on Walker-walk with Walker-stand pretrained for 60K steps, with four convolutional layers frozen. We provide analysis for the number of layers frozen and number of environment interactions before freezing $T_f$ in Figure 9. We find that freezing more layers allows for more computational gain, since we can avoid computing gradients for the frozen layers without sacrificing performance. Longer pretraining in the source task improves compute-efficiency in the target task; however, early convergence of encoder parameters enables the agent to learn a good policy even with only 20K interactions before transfer.

We remark that Yosinski et al. (2014) examine the generality of features learned by neural networks and the feasibility of transferring parameters between similar image classification tasks. Yarats et al. (2019) show that transferring encoder parameters pretrained from Walker-walk to Walker-stand and Walker-run can improve the performance and sample-efficiency of a SAC agent. For the first time, we show that encoder parameters trained on simple tasks can be useful for computation-efficient training in complex tasks and new domains.

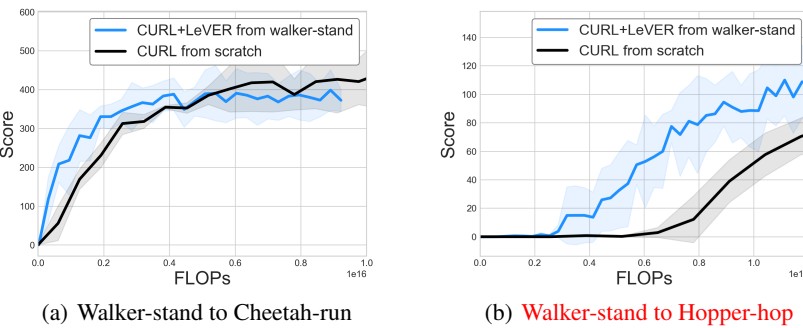

(a) Walker-stand to Cheetah-run      (b) Walker-stand to Hopper-hop

Figure 8: Comparison of the computational efficiency of agents trained from scratch with CURL and agents trained with CURL+LeVER from Walker-stand pretraining.

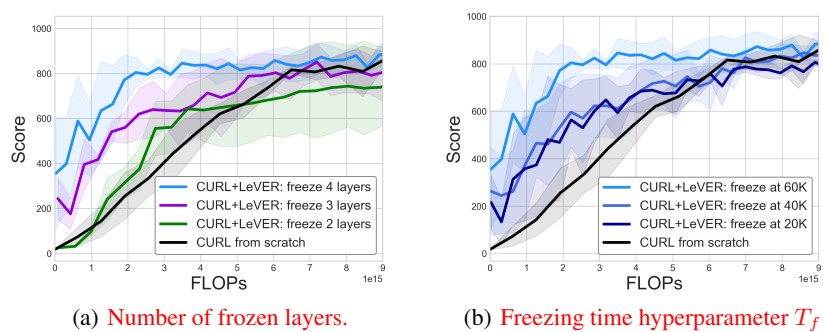

(a) Number of frozen layers.      (b) Freezing time hyperparameter $T_f$.

Figure 9: (a) Analysis on the number of frozen convolutional layers in Walker-walk training from Walker-stand pretrained for 60K steps. (b) Analysis on the number of environment steps Walker-stand agent is pretrained prior to Walker-walk transfer, where the first four convolutional layers are frozen.

## D   COMPUTATIONAL EFFICIENCY IN CONSTRAINED-MEMORY SETTINGS

In our main experiments, we isolate the two major contributions of our method, reduced computational overhead and improved sample-efficiency in constrained-memory settings. In Figures 10 and 11 we show that these benefits can also be combined for significant computational gain in constrained-memory settings.

## E   SAMPLE-EFFICIENCY PLOTS

In section 5.2 we show the computational efficiency of our method in DMControl and Atari environments. We show in Figure 12 that our sample-efficiency is very close to that of baseline CURL (Srinivas et al., 2020), with only slight degradation in Cartpole-swingup and Walker-walk. In Atari games (Figure 13), we match the sample-efficiency of baseline Rainbow (Hessel et al., 2018) very closely, with no degradation.

## F   GENERAL IMPLEMENTATION DETAILS

LeVER can be applied to any convolutional encoder which compresses the input observation into a latent vector with smaller dimension than the observation. We generally freeze all the convolutional

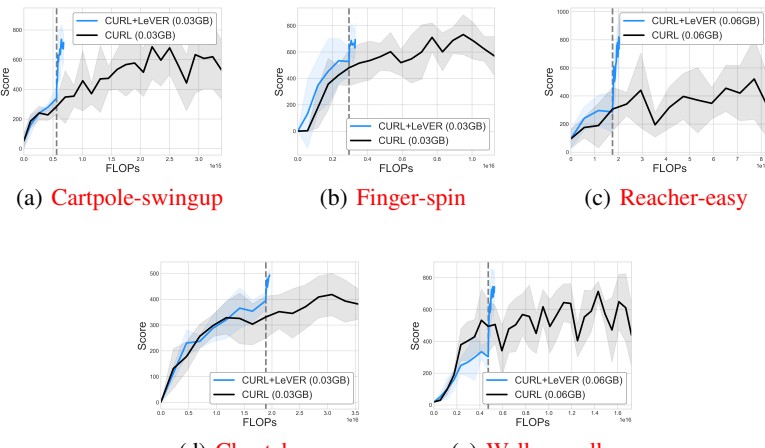

Figure 10: Comparison of CURL in constrained-memory settings with and without LeVER, where the x-axis shows FLOPs, corresponding to Figure 5. The dotted gray line denotes the encoder freezing time $t = T_f$. The solid line and shaded regions represent the mean and standard deviation, respectively, across five runs.

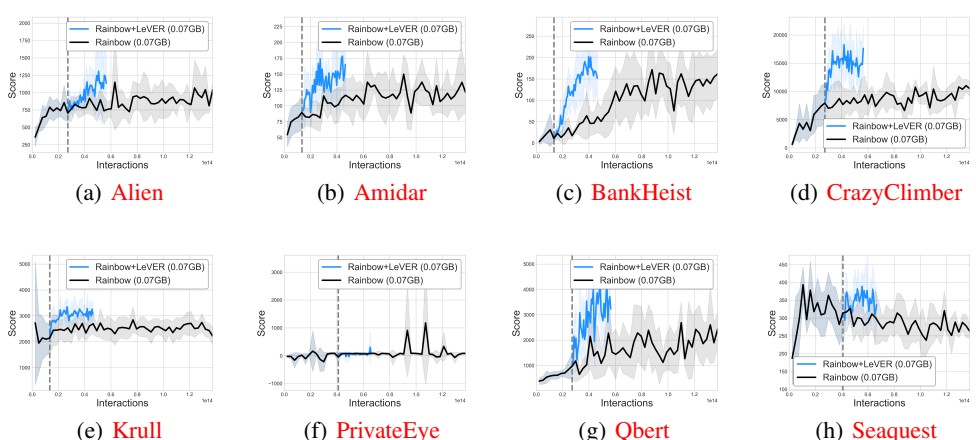

Figure 11: Comparison of Rainbow in constrained-memory settings with and without LeVER, where the x-axis shows FLOPs, corresponding to Figure 4. The dotted gray line denotes the encoder freezing time $t = T_f$. The solid line and shaded regions represent the mean and standard deviation, respectively, across five runs.

layers and possibly the first fully-connected layer. In our main experiments, we chose to freeze the first fully-connected layer for DM Control experiments and the last convolutional layer for Atari experiments. We made this choice in order to simultaneously save computation and memory; for those architectures, if we freeze an earlier layer, we save less computation, and the latent vectors (convolutional features) are too large for our method to save memory. In DM Control experiments, the latent dimension of the first fully-connected layer is 50, which allows a roughly 12X memory

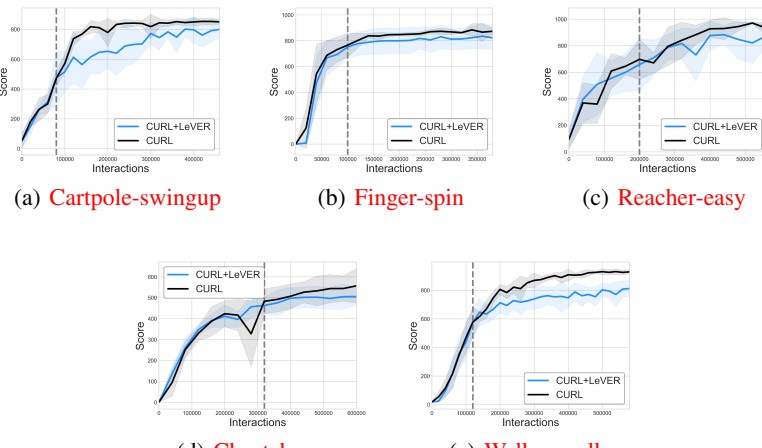

Figure 12: Comparison of the sample-efficiency of CURL with and without LeVER, corresponding to Figure 2. The dotted gray line denotes the encoder freezing time $t = T_f$. The solid line and shaded regions represent the mean and standard deviation, respectively, across five runs.

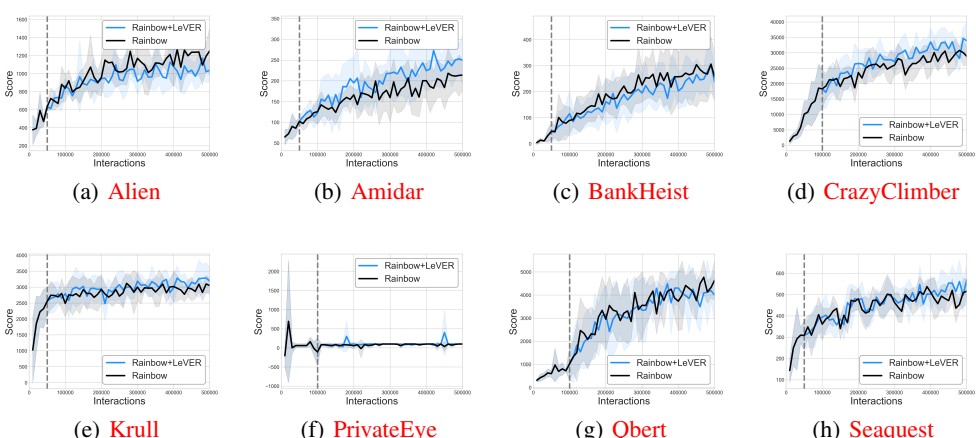

Figure 13: Comparison of the sample-efficiency of Rainbow with and without LeVER, corresponding to Figure 3. The dotted gray line denotes the encoder freezing time $t = T_f$. The solid line and shaded regions represent the mean and standard deviation, respectively, across five runs.

gain. In Atari experiments, the latent dimension of the last convolutional layer is 576, which allows a roughly 3X memory gain.

## G  DMCONTROL IMPLEMENTATION DETAILS

We use the network architecture in https://github.com/MishaLaskin/curl for our CURL (Srinivas et al., 2020) implementation. We show a full list of hyperparameters in Table 4.

## H  ATARI IMPLEMENTATION DETAILS

We use the network architecture in https://github.com/Kaixhin/Rainbow for our Rainbow (Hessel et al., 2018) implementation and the data-efficient Rainbow (van Hasselt et al., 2019) encoder architecture and hyperparameters. We show a full list of hyperparameters in Table 5.

Table 4: Hyperparameters used for DMControl experiments. Most hyperparameter values are unchanged across environments with the exception of initial replay buffer size, action repeat, and learning rate.

| Hyperparameter | Value |
|---|---|
| Augmentation | Crop |
| Observation rendering | $(100, 100)$ |
| Observation down/upsampling | $(84, 84)$ |
| Replay buffer size in Figure 2 | Number of training steps |
| Initial replay buffer size in Figure 5 | 1000 cartpole, swingup; cheetah, run; finger, spin |
| | 2000 reacher, easy; walker, walk |
| Number of updates per training step | 1 |
| Initial steps | 1000 |
| Stacked frames | 3 |
| Action repeat | 2 finger, spin; walker, walk |
| | 4 cheetah, run; reacher, easy |
| | 8 cartpole, swingup |
| Hidden units (MLP) | 1024 |
| Evaluation episodes | 10 |
| Evaluation frequency | 2500 cartpole, swingup |
| | 10000 cheetah, run; finger, spin; reacher, easy; walker, walk |
| Optimizer | Adam |
| $(\beta_1, \beta_2) \rightarrow (f_\psi, \pi_\phi, Q_\theta)$ | $(.9, .999)$ |
| $(\beta_1, \beta_2) \rightarrow (\alpha)$ | $(.5, .999)$ |
| Learning rate $(f_\psi, \pi_\phi, Q_\theta)$ | $2e - 4$ cheetah, run |
| | $1e - 3$ cartpole, swingup; finger, spin; reacher, easy; walker, walk |
| Learning rate $(\alpha)$ | $1e - 4$ |
| Batch Size | 512 cheetah, run |
| | 128 cartpole, swingup; finger, spin; reacher, easy; walker, walk |
| $Q$ function EMA $\tau$ | 0.01 |
| Critic target update freq | 2 |
| Convolutional layers | 4 |
| Number of filters | 32 |
| Non-linearity | ReLU |
| Encoder EMA $\tau$ | 0.05 |
| Latent dimension | 50 |
| Discount $\gamma$ | .99 |
| Initial temperature | 0.1 |
| Freezing time $T_f$ in Figure 2 | 10000 cartpole, swingup |
| | 50000 finger, spin; reacher, easy |
| | 60000 walker, walk |
| | 80000 cheetah, run |
| Freezing time $T_f$ in Figure 5 | 10000 cartpole, swingup |
| | 50000 finger, spin |
| | 30000 reacher, easy |
| | 80000 cheetah, run; walker, walk |

Table 5: Hyperparameters used for Atari experiments. All hyperparameter values are unchanged across environments with the exception of encoder freezing time.

| Hyperparameter | Value |
| --- | --- |
| Augmentation | None |
| Observation rendering | $(84, 84)$ |
| Replay buffer size in Figure 3 | Number of training steps |
| Initial replay buffer size in Figure 4 | 10000 |
| Number of updates per training step | 1 |
| Initial steps | 1600 |
| Stacked frames | 4 |
| Action repeat | 1 |
| Hidden units (MLP) | 256 |
| Evaluation episodes | 10 |
| Evaluation frequency | 10000 |
| Optimizer | Adam |
| $(\beta_1, \beta_2) \rightarrow (f_\psi, Q_\theta)$ | $(.9, .999)$ |
| Learning rate $(f_\psi, Q_\theta)$ | $1e - 3$ |
| Learning rate $(\alpha)$ | 0.0001 |
| Batch Size | 32 |
| Multi-step returns length | 20 |
| Critic target update freq | 2000 |
| Convolutional layers | 2 |
| Number of filters | $32, 64$ |
| Non-linearity | ReLU |
| Discount $\gamma$ | .99 |
| Freezing time $T_f$ in Figure 3 | 50000 Alien; Amidar; BankHeist; Krull; Seaquest |
|  | 100000 CrazyClimber; PrivateEye; Qbert |
| Freezing time $T_f$ in Figure 4 | 50000 Amidar; BankHeist; Krull |
|  | 100000 Alien; CrazyClimber; Qbert |
|  | 150000 PrivateEye; Seaquest |

