# OpenReview forum: "Compute- and Memory-Efficient Reinforcement Learning with Latent Experience Replay"
_ICLR.cc/2021/Conference — Reject_

### Official Review · AnonReviewer1 · 2020-10-28
**Simple idea, good manuscript.**

**Rating:** 7
**Confidence:** 4

**Review:**

### Paper summary

This work proposes LeVER, a method that modifies general off-policy RL algorithms with a fixed layer freezing policy for early embedding layers (in this particular case, a few early layers of a CNN). As a direct consequence, the method enables to store embeddings in the experience replay buffer rather than observations, with a potential decrease in memory required, as well as providing a boost in clock time due to fewer gradient computations needed for every update. The method is benchmarked with a couple of off-policy RL algorithms against a few different environments.

### Good things

- The approach is extremely relevant to most of the RL community. We are training for longer periods of time and generating significantly more data than we did a few years ago, so any method that enables to increase training efficiency is extremely welcomed.
- The method is simple, but clever, and the manuscript quite nicely details the steps taken to improve learning stability (which can arise due to the obvious possibility of bad model freezes).
- The coverage of related work in the literature throughout the manuscript is excellent, and provides enough pointers for the reader to understand how the manuscript is placed in it.
- The experimental setting clearly states hypotheses and questions that will be answered.
- Section 5.4 convincingly argues that the freezing method is empirically justified.

### Concerns

This is a good paper, so generally I don't have any strong negative thoughts, however I think it would be good to report how the method does when different choices are made with respect to how much of the network is frozen.
That is, in the proposed experiment setting the choices were reasonable but nonetheless a little arbitrary, so knowing a little bit more about learning dynamics with this approach would probably make the paper stronger and more robust for future readers.

### Questions:

- I wonder whether the authors would shed more details on the transfer learning setting (e.g. whether the transfer capacity changes wrt. changes in method hyperparameters such as freezing time, different saved embedding functions, etc.), and whether the results do generally show up in more environments/algorithms.
- The reduction in variance after the freezing is interesting; I wonder if the plots could show all the single runs, and whether the authors have any explanations for this somewhat consistent (!) change in learning behaviour.

---

> ### Author Response · Authors · 2020-11-18
> **Response to Reviewer 1**
>
> Thank you for your feedback and positive assessment of our paper! As you, R3, and R4 mention, our work is well-motivated and has significance to the RL community. We introduce a simple yet effective method for compute- and memory-efficient RL. Our careful implementation considerations and extensive experimental results are important technical contributions which allow the method to be widely adopted.
>
> We have revised our draft based on your suggestions. Revised parts in the new draft are colored red (in particular, we updated Sections 2 and 5, Appendices B, C, and F, and Figures 2, 5, 6, 7a, 8, 9, 10, and 12. These refer to sections and figures in the revised version, as the ordering may have changed.). We address your comments and questions below:
>
> ---
> **Q1: Clarification and more ablations on freezing**
>
> A1: In our main experiments, we chose to freeze the first fully-connected layer for DM Control experiments and the last convolutional layer for Atari experiments. We made this choice in order to simultaneously save computation and memory; for those architectures, if we freeze an earlier layer, we save less computation, and the latent vectors (convolutional features) are too large for our method to save memory.
>
> Following your suggestion, we did an ablation on the number of layers frozen and an ablation on the source task freezing time for Walker-stand to Walker-walk in Appendix C. We found that freezing more layers allows for more computational gain, since we can avoid computing gradients for the frozen layers without sacrificing performance. Longer pretraining in the source task improves compute-efficiency in the target task; however, early convergence of encoder parameters enables the agent to learn a good policy even with only 20K interactions before transfer. We plan to add a similar ablation for Figure 2, time permitting, but we believe our added ablation demonstrates the trend.
>
> ---
> **Q2: More details and results on transfer learning setting**
>
> A2: We have added more transfer experiments in the revised draft to better explain the transfer setting. We added two ablations for Walker-stand to Walker-walk in Appendix C (more details in our answer above). We also added a domain transfer experiment for Walker-stand to Hopper-hop in Appendix C and conclude that these results do show up across multiple environments.
>
> ---
> **Q3: Reduction in variance**
>
> A3: We hypothesize that our method stabilizes individual runs since only the Q-function and policy MLPs (which are usually modeled by a few MLP layers) are updated and the parameters are changing less overall. However, we have not investigated this behavior further and are not certain that variance between runs is reduced. Could you point out which figure you would like to see single runs for?
>
> ---
> Thank you for your suggestions to improve the clarity and robustness of our paper! We hope to have addressed all of your questions.

---

### Official Review · AnonReviewer2 · 2020-10-28
**A Memory and compute optimization method that replaces an image based replay buffer with a buffer that stores low dimensional learned representations.**

**Rating:** 5
**Confidence:** 5

**Review:**

Significance:
The paper proposes to reduce memory and computation demands of an image based RL by exploiting early convergence of the convolutional encoder. While the approach is quite intriguing, I find it hard to see the approach being general and thus having a significant effect on the RL community.

Pros:
The paper provides an interesting approach in order to save memory and computational footprint in image-based deep RL. The method is based on an observation that the convolutional encoder converges faster then the actor and critic networks.
The authors provide an extensive comparison and ablation study that covers different domains (DMControl and Atari). The ablation study shades more light on some of the properties for the training dynamics of an image based model free RL algorithm (attention map, layer convergence)
Cons:
I’m a bit skeptical about the generality of this approach. Freezing the encoder’s weights prematurely prevents the encoder to adequately encode priorly unseen images (out of distribution), this in turn will hurt the agent’s performance down the road. Note that in DMControl the method is only showcased on the simplest tasks (at least cheetah run should be included) where learning a good policy only takes about 100K env steps -- enough to collect sufficient data support to learn an almost optimal policy. Figure 6c adheres to my point here, as conv1 weights pretty much don’t change, suggesting convergence.
The task transfer (walker stand to walker walk) experiment is exactly the same as the one demonstrated in SAC-AE (https://arxiv.org/pdf/1910.01741.pdf, Section 5.3). I’m not sure what is the difference here besides using CURL instead of SAC-AE. Could the authors elaborate? Also the domain transfer experiment (App G) shows that the approach doesn’t really buy anything.
It seems that the approach requires storing 4 data augmented observations per env observation and the replay buffer size is equal to the number of training steps. I would like to point out that DrQ (https://arxiv.org/pdf/2004.13649.pdf) only needs a constant size replay buffer of 100K transitions, even if training requires 3M steps. Given that, I’m skeptical that LeVER would buy much in terms of memory and computation here. It would be nice to see a head to head comparison.
Results are demonstrated over 3 random seeds, which is too few to get any conclusive statistical evidence given the variance. A common practice is to use 10 seeds for DMControl and 5 seeds for Atari.


Quality:
While the technical contributions of the paper are limited in novelty and significance, and don’t meet the high acceptance bar of ICLR, I still think the paper is well done and could be a good workshop paper.


Clarity:
The paper in general is clearly written and well organized. I particularly appreciate the contribution bullet points and the experimentation roadmap.

------------------------------------------
Post rebuttal:

The authors has addressed several of my concerns regarding the method's generality and some experiments. While I'm raising my score to 5, I'm still not convinced that the paper proposed a valuable contribution to the community -- comparing RL algorithms in memory or compute footprints instead of the number interactions with the environment is not meaningful, especially when a simulator is in use. There are several much simpler things one can do to tradeoff compute or memory (for example re-render observations on the fly from stored low dimensional states). Thus, I'm voting for a rejection.

---

> ### Author Response · Authors · 2020-11-18
> **Response to Reviewer 2**
>
> Thank you for the helpful and valuable feedback on our paper! As R1, R3, and R4 mention, our work is well-motivated and has significance to the RL community. We introduce a simple yet effective method for compute- and memory-efficient RL. Our careful implementation considerations and extensive experimental results are important technical contributions which allow the method to be widely adopted. We will release our implementation so that it is useful to everyone.
>
> We have revised our draft based on your suggestions. Revised parts in the new draft are colored red (in particular, we updated Sections 2 and 5, Appendices B, C, and F, and Figures 2, 5, 6, 7a, 8, 9, 10, and 12. These refer to sections and figures in the revised version, as the ordering may have changed.). We address your comments and questions below:
>
> ---
> **Q1: Generality of our method**
>
> A1: To address your concerns about Cheetah-run, we have added Cheetah-run experiments in Figure 2, where CURL+LeVER achieves better compute-efficiency, and Figure 5, where CURL+LeVER outperforms CURL when both are restricted to a replay capacity of 1K. We also remark that we added more experiments (Figure 6b) using a different encoder architecture (i.e. IMPALA network from Espeholt et al., 2018). We believe that our extensive experiments on various domains (including Atari) and setups show the generality of the proposed method. Note that we carefully tune the freezing time hyperparameter to train for enough time to avoid premature freezing, but before the agent learns a good policy.
>
> In terms of encoding priorly unseen images, the encoder learns generalizable features which are relevant even for images it has never seen before. In fact, in our domain transfer experiments, the encoder has never been trained on Cheetah-run or Hopper-hop (added in revised version), but it is reusable in these unseen environments. In these cases, the Cheetah-run and Hopper-hop observations are out-of-distribution, but the frozen encoder trained on Walker-stand can capture the important aspects for control.
>
> Overall, we show the generality of our method with experimental results across 5 DM Control and 8 Atari environments, 2 domain transfer settings, and 2 very different encoder architectures.
>
> ---
> **Q2: Similarity to task transfer experiment in SAC-AE**
>
> A2: Thank you for pointing out this task transfer experiment; we have added a citation to this experiment in Section 5.5 and Appendix C. However, we remark that the main purpose of our experiments is a little bit different from that of SAC-AE. Our work emphasizes compute-efficiency, while previous works largely focus on sample-efficiency. We also remark that we pretrain on Walker-stand, which is an easier task than Walker-walk, and we consider generalization to unseen domains, such as Cheetah and Hopper.
>
> ---
> **Q3: Domain transfer effectiveness**
>
> A3: We have added a domain transfer experiment for Walker-stand to Hopper-hop in Figure 8b, which better showcases the compute-efficiency of our method than the Walker-stand to Cheetah-run experiment.
>
> ---
> **Q4: Comparison to DrQ constant size replay buffer**
>
> A4: We used a buffer size equal to the number of training steps in our compute-efficiency plots in Figures 2 and 3 in order to focus solely on the computational gain from LeVER. However, this is not a requirement for LeVER and there is no reason that we cannot use 100K initial replay capacity. In fact, we show in our memory-efficiency plots (Figures 4 and 5) that LeVER can perform better than baseline methods when memory is limited, so constraining the replay memory would actually show more benefits of LeVER!
>
> ---
> **Q5: Number of random seeds**
>
> A5: We plan to increase the number of random seeds to 5 in our main experiments by the end of the rebuttal period and to add 5 more seeds for DM Control experiments and seeds for the ablations in the camera-ready version due to time constraints of the rebuttal period.
>
> ---
> Thank you for your suggestions to improve the clarity and robustness of our paper! We hope to have addressed your concerns.

---

### Official Review · AnonReviewer3 · 2020-10-28
**Official Blind Review #3**

**Rating:** 6
**Confidence:** 4

**Review:**

- Summary
    - This paper presents a method for compute- and memory-efficient reinforcement learning where the visual encoder is frozen partway into training.  After freezing latent vectors are stored in the replay buffer instead of images (and any existing images are replaced by them).  This leads to both better compute and memory utilization.
    - The authors demonstrate their method by comparing to Rainbow on Atari and CURL on DM Control.  On DM control, their method reduces the compute by a considerable margin.  On Atari, the results are less clear cut, but the compute cost is reduced.
    - When they also impose a memory constraint, the effectiveness of their method is further increased.
- Strengths
    - Elegant and "obvious in hindsight" (a good thing) idea, meaning it will likely have broad applicability
    - While the authors only tested it on off-policy methods, it is clearly also applicable to on-policy methods that use a rollout storage (PPO, PPG, IMPALA, V-MPO, A2C, etc.)
    - Good FLOPs vs. Reward results on DM Control
- Weaknesses
    - The memory constrained results seem very contrived.  60 MB is a tiny amount of memory and even 9.0 GB of (presumably) CPU memory isn't that prohibitive.
        - Perhaps if wall-clock time was plotted in addition to samples, the smaller memory footprint of LeVER would mean the replay buffer can be stored on the GPU and training would be much faster since many expensive CPU -> GPU transfers would be eliminated?
    - The CNN is frozen all at once instead of frozen iteratively. Raghu 2017 and Figure 6c suggest that the early layers could be frozen much earlier, although this may increase the memory usage initially since CNNs typically increase the memory size of the feature map in lower layers.
    - T_f seems like yet another hyper-parameter to tune.  In theory, SVCCA (or PWCCA from Morcos 2018) could be used to choose when to freeze (if the representation of the shallowest unfrozen layer didn't change in the last K steps, freeze it).  There is a nontrivial cost to computing either of those so it
- Suggestions for improvement
    - I very much like the idea of this paper, but I think the chosen application is making the idea look less convincing (i.e. freezing the CNN isn't really that impactful when the CNN and observations are tiny).  I urge the authors to try this for visual navigation (i.e. PointGoal Navigation in Habitat/AI2 Thor/etc), where deeper CNNs, e.g. ResNet18, and higher resolution images, e.g. 256x256, are used.
    - One other potentially benefit of LeVER is the ability to increase the batch size during training (as in Smith 2017).  This could perhaps increase its effectiveness further?
    - In figure 6a, there should also be a CURL + LeVER from Scratch line.  Currently two variables are changing.
    - One paper that should be cited is Fang 2019.  They do a very similar thing as LeVER out of necessity.
- Overall
    - While I like this paper and think the idea has a lot of potential, I don't think it is quite ready for publication yet.  I urge the authors to try their idea in a setting with a larger CNN and higher resolutions and to see if there is a way to find T_f without it being "yet another hyper-parameter".
- References
    - Smith 2017: https://arxiv.org/abs/1711.00489
    - Morcos 2018: https://arxiv.org/abs/1806.05759
    - Fang 2019: https://arxiv.org/abs/1903.03878

## Post Rebuttal

I thank the authors for their responses. The results with a larger CNN and increased batch size help show the benefit of the method further.  I still believe the presentation of the method would be considerably stronger if results were presented in a setting with larger CNNs and higher resolution.

---

> ### Author Response · Authors · 2020-11-18
> **Response to Reviewer 3**
>
> Thank you for the helpful and valuable feedback on our paper! As you, R1, and R4 mention, our work is well-motivated and has significance to the RL community. We introduce a simple yet effective method for compute- and memory-efficient RL. Our careful implementation considerations and extensive experimental results are important technical contributions which allow the method to be widely adopted.
>
> We have revised our draft based on your suggestions. Revised parts in the new draft are colored red (in particular, we updated Sections 2 and 5, Appendices B, C, and F, and Figures 2, 5, 6, 7a, 8, 9, 10, and 12. These refer to sections and figures in the revised version, as the ordering may have changed.).  We address your comments and questions below:
>
> ---
> **Q1: Memory constraints aren't prohibitive**
>
> A1: In use-cases like mobile robots where learning is asynchronous and completely on-device, 9GB of CPU RAM can be prohibitive or consume significant amounts of battery power. If the user wants to train for longer, the amount of memory required is even larger. We agree that another potential benefit is the ability to store the replay buffer in GPU and reduce expensive CPU to GPU transfers and think this would be very interesting future work!
>
> ---
> **Q2: Freezing layers iteratively**
>
> A2: In deep RL, the convolutional networks are usually shallower than those used in supervised learning and other domains, so we find that freezing the encoder layers all at once provides a considerable compute gain already. However, we think that freezing the layers iteratively should be an interesting future direction to explore.
>
> ---
> **Q3: Choosing $T_f$ with SVCCA or PWCCA**
>
> A3: We have considered using network similarity metrics (like SVCCA and PWCCA) to determine the freezing time, but found we would still need to tune a task-specific hyperparameter for the similarity threshold. As you mention, there is also a nontrivial cost of computing these similarity scores, so given that these metrics still introduces a hyperparameter, we believe it makes more sense to directly tune $T_f$.
>
> ---
> **Q4: Results with larger CNNs and observations**
>
> A4: Thank you for the suggestion to show our method more convincingly! We evaluate our method in DM Control and Atari because they are common RL benchmarks used in many recent works on RL from pixels (CURL, SAC-AE, SimPLe and so on). However, we agree that larger CNNs and higher resolution observations would show our results more effectively. To this end, we have added experiments with the IMPALA encoder architecture (see Section 5.3), a deeper convolutional network containing residual blocks. The results indeed show a larger computational saving from our method. Due to time constraints of the rebuttal period, we cannot prepare results in new environments during this period, but we plan to prepare PointGoal Navigation in Habitat experiments for the camera-ready version.
>
> ---
> **Q5: Increasing the batch size**
>
> A5: Thank you for pointing out this additional benefit of LeVER! It has been observed that larger batch sizes are useful for methods such as CURL and RAD, but increase computational overhead. LeVER enables using larger batch sizes without incurring high computational cost. We added an experiment comparing the sample-efficiency and compute-efficiency of CURL (b=128) and CURL+LeVER (b=512) in Section 5.4. We found that in Cheetah-run, CURL+LeVER (b=512) has better sample-efficiency than CURL (b=128), but uses similar computational resources. We plan to add a CURL (b=512) learning curve to Figure 6a soon. As you mention, it could also be possible to adaptively increase the batch size after freezing; we will leave this for future work!
>
> ---
> **Q6: CURL+LeVER from scratch line**
>
> A6: Thank you for pointing this out! We have added a CURL+LeVER from scratch line to Figure 7a. We find that CURL+LeVER with pre-training is more compute-efficient than CURL+LeVER from scratch since the encoder parameters are not updated at all during training on the target task. We will add a CURL+LeVER from scratch line to our domain transfer experiments for the camera-ready version.
>
> ---
> **Q7: Related work**
>
> A7: Thank you for sharing “Scene Memory Transformer for Embodied Agents in Long-Horizon Tasks” with us; we have added it to our related work in Section 2.
>
> ---
> Thank you for your suggestions to improve the clarity and robustness of our paper! We hope to have addressed your concerns.

---

### Official Review · AnonReviewer4 · 2020-11-02
**The experimental results show impressive improvement, but the proposed technique lacks rigorous definition and explanation.**

**Rating:** 6
**Confidence:** 3

**Review:**

This manuscript proposes to reduce the intensive computation and memory requirement in reinforcement learning trainings by freezing the parameters of lower layers early. Besides, the authors also propose to store the low-dimensional latent vectors rather than the high-dimensional images in the replay buffer for experience replay. The effectiveness of the proposed techniques is evaluated on DeepMind Control environments and Atari. The motivation for this work is strong, and the results are impressive. However, the proposed technique is described in a very general way without clearly defined applicable conditions and specific design methods. Below are detailed comments and questions.

1: the main idea is to freeze lower layers of CNN encoders. However, for a certain network with various structures, is there any applicable conditions or limitations? How to choose the number of layers to freeze? The proposed technique needs to rigorous definition and explanation.

2: It seems the reduction comes from the freezing of lower layers. I am wondering what is the computation/memory requirement breakdown in terms of layers? Is it always the case that lower layers consume a significant amount of computation and memory? If it is not the case, can LeVER still be effective?

3: this paper also proposes to store latent vectors instead of high-dimensional images. There again lacks detailed description and explanation of the applicable conditions and to what extent we can reduce the dimension of the latent vectors.

---

> ### Author Response · Authors · 2020-11-18
> **Response to Reviewer 4**
>
> Thank you for the helpful and valuable feedback on our paper! As you, R1, and R3 mention, our work is well-motivated and has significance to the RL community. We introduce a simple yet effective method for compute- and memory-efficient RL. Our careful implementation considerations and extensive experimental results are important technical contributions which allow the method to be widely adopted.
>
> We have revised our draft based on your suggestions. Revised parts in the new draft are colored red (in particular, we updated Sections 2 and 5, Appendices B, C, and F, and Figures 2, 5, 6, 7a, 8, 9, 10, and 12. These refer to sections and figures in the revised version, as the ordering may have changed.).  We address your comments and questions below:
>
> ---
> **Q1: Rigorous definition of our technique**
>
> A1: Thank you for the suggestion to make our paper more clear and define our technique more rigorously! We added Appendix F to explain how we choose the number of layers to freeze. In our main experiments, we chose to freeze the first fully-connected layer for DM Control experiments and the last convolutional layer for Atari experiments. We made this choice in order to simultaneously save computation and memory; for those architectures, if we freeze an earlier layer, we save less computation, and the latent vectors (convolutional features) are too large for our method to save memory. Additionally, we added an ablation in Appendix C to show the compute-efficiency for various numbers of frozen layers for our task transfer setup and found that freezing more layers allows for more computational gain.
>
> In terms of applicable conditions or limitations, we believe LeVER can be applied to most convolutional encoders which compress image observations into latent vectors before passing them into Q-function/policy networks. To show the applicability of LeVER to different network architectures, we added experiments with the IMPALA encoder in Section 5.3 to show that LeVER can also be used for larger architectures with residual connections.
>
> ---
> **Q2: Computation / memory breakdown by layer**
>
> A2: We have added a FLOPs breakdown by layer for our experiments in Appendix B, although this is architecture-specific. In pixel-based RL, it is always the case that lower layers consume significant computation, since the convolutional encoder compresses the image observation into low-dimensional latent vectors.
>
> ---
> **Q3: Reducing dimension of latent vectors**
>
> A3: To what extent we can reduce the dimension of the latent vectors depends on the network architecture. Typically, the convolutional encoder will compress the image observation into much smaller latent vectors. For example, the architecture we use in DM Control compresses a 3x84x84 image to a size 50 latent vector; the architecture we use in Atari compresses a 84x84 image to a size 576 latent vector. The applicable condition is that the latent vector should have smaller dimensionality than the input observation, after accounting for the additional factors explained in Section 4.2. Most modern convolutional neural networks (e.g., ResNet, VAEs, etc.) have this property because they project the image to a low-dimensional latent vector.
>
> ---
> Thank you for your suggestions to improve the clarity and robustness of our paper! We hope to have addressed all of your questions.

---

### Author Response · Authors · 2020-11-18
**Common response to all reviewers: short summary of rebuttal**

Dear reviewers,

We really appreciate your efforts in providing insightful comments on our manuscript.

To best respond to your questions and concerns, we have carefully revised and enhanced our manuscript with a substantial amount of additional experiments following suggestions from your comments, including:

 - Compute- and memory-efficiency of LeVER in Cheetah-run (Figures 2 and 5)
 - Compute- and sample-efficiency of LeVER with large batch sizes (Figure 6a)
 - Compute-efficiency of LeVER with IMPALA architecture (Figure 6b)
 - Compute-efficiency of LeVER on Walker-stand to Hopper-hop transfer (Figure 8b)
 - Ablations on number of frozen layers and freezing time for task transfer (Figure 9)

Revised parts in the new draft are colored red (in particular, we updated Sections 2 and 5, Appendices B, C, and F, and Figures 2, 5, 6, 7a, 8, 9, 10, and 12. These refer to sections and figures in the revised version, as the ordering may have changed.).

Thank you very much!
Authors

---

### Author Response · Authors · 2020-11-22
**After first revision**

Dear reviewers,

We sincerely appreciate your time and effort to review our paper.

Since the second discussion phase will end soon, please let us know if you have any comments/concerns that we have not addressed up to your satisfactory. We will be happy to clarify further and strengthen our paper.

Thank you very much!

Authors

---

### Author Response · Authors · 2020-11-24
**Additional updates after first revisions**

Dear reviewers,

We have made more improvements to the manuscript. Changes are highlighted in red. Please let us know if you have any comments/concerns we have not addressed!

In particular,
- We added a CURL (b=256) line to Figure 6a. CURL+LeVER (b=512) achieves similar sample-efficiency to CURL (b=256) while consuming significantly less compute (similar compute-efficiency to CURL (b=128)). We previously mentioned adding CURL (b=512), but we felt that b=256 sufficiently demonstrated this message.
- We have increased the number of random seeds to 5 in all of our main results (Figures 2, 3, 4, and 5).

Thank you again for providing us with feedback to strengthen our paper!

Authors.

---

### Decision · Program_Chairs · 2021-01-07
**Final Decision**

**Decision:**

Reject

**Comment:**


This paper presents approach to improve compute and memory efficiency by freezing layers and storing latent features. The approach is simple and provide efficiency. However, there are concerns as well. One big concern is that the experiments are not on realistic settings for example real world images and the current CNN is too simple. Overall, the reviewers are split. The AC agrees with some of the reviewers that for a paper like this experiments on more realistic setting will make it significantly stronger.